foxr1 is a novel maternal-effect gene in fish that is required for early embryonic success

Cheung Caroline T.
Patinote Amélie
http://orcid.org/0000-0001-5464-6219 Guiguen Yann
http://orcid.org/0000-0002-9355-8227 Bobe Julien Julien.Bobe@inra.fr
LPGP, UR1037 Fish Physiology and Genomics, INRA , Rennes , France
Posner Mason
Electronic publication date: 2018 Aug 23
Publication date: 2018
Volume: 6
Electronic Location ID: e5534
Received 2018 Apr 11; Accepted 2018 Aug 8
Copyright: © 2018 Cheung et al.
Copyright year: 2018
Copyright holder: Cheung et al.
License: This is an open access article distributed under the terms of the Creative Commons Attribution License, which permits unrestricted use, distribution, reproduction and adaptation in any medium and for any purpose provided that it is properly attributed. For attribution, the original author(s), title, publication source (PeerJ) and either DOI or URL of the article must be cited.
License URL: https://creativecommons.org/licenses/by/4.0/

Keywords: CRISPR-cas9, Maternal-effect genes, foxr1, p21, Cell growth and survival, Rictor, Zebrafish, Egg quality

Funding: French National Research Agency ANR Maternal Legacy ANR-13-BSV7-0015 This work was supported by the French National Research Agency ANR (Maternal Legacy ANR-13-BSV7-0015) and by Région Bretagne. The funders had no role in study design, data collection and analysis, decision to publish, or preparation of the manuscript.

==============================
The family of forkhead box (Fox) transcription factors regulates gonadogenesis and embryogenesis, but the role of foxr1 in reproduction is unknown. Evolutionary history of foxr1 in vertebrates was examined and the gene was found to exist in most vertebrates, including mammals, ray-finned fish, amphibians, and sauropsids. By quantitative PCR and RNA-seq, we found that foxr1 had an ovarian-specific expression in zebrafish, a common feature of maternal-effect genes. In addition, it was demonstrated using in situ hybridization that foxr1 was a maternally-inherited transcript that was highly expressed even in early-stage oocytes and accumulated in the developing eggs during oogenesis. We also analyzed the function of foxr1 in female reproduction using a zebrafish CRISPR/cas9 knockout model. It was observed that embryos from the foxr1-deficient females had a significantly lower survival rate whereby they either failed to undergo cell division or underwent abnormal division that culminated in growth arrest at around the mid-blastula transition and early death. These mutant-derived eggs contained dramatically increased levels of p21, a cell cycle inhibitor, and reduced rictor, a component of mTOR and regulator of cell survival, which were in line with the observed growth arrest phenotype. Our study shows for the first time that foxr1 is an essential maternal-effect gene and may be required for proper cell division and survival via the p21 and mTOR pathways. These novel findings will broaden our knowledge on the functions of specific maternal factors stored in the developing egg and the underlying mechanisms that contribute to reproductive success.

Introduction

In vertebrates, maternal products including transcripts, proteins, and other biomolecules are necessary for initiating early embryonic development from fertilization until the mid-blastula transition (MBT) when the zygotic genome is activated (Baroux et al., 2008). Maternal-effect genes are transcribed from the maternal genome and encode the maternal factors that are deposited into the developing oocytes in order to coordinate embryonic development before MBT (Lindeman & Pelegri, 2010). We had previously explored the zebrafish egg transcriptome (Cheung et al., 2018) and proteome (Yilmaz et al., 2017) in order to gain further understanding of the maternal factors that contribute to good quality or developmentally competent eggs that result in high survival of progeny. However, despite the increasing identification of maternal-effect genes and their functions, large gaps still remain especially the role of genes that regulate early embryogenesis.

The forkhead box (Fox) proteins belong to a family of transcription factors that play important roles in cell growth, proliferation, survival, and cell death (Hannenhalli & Kaestner, 2009). Many of these Fox proteins have been shown to be essential to the various processes of embryogenesis. In mammals, knockouts of several fox genes, including foxa2, foxo1, and foxf1, result in embryonic lethality due to defects in development of different organs (Hannenhalli & Kaestner, 2009; Martins, Lithgow & Link, 2016; Mahlapuu et al., 2001). In reproduction, a recent transcriptomic study in the Nile tilapia, Oreochromis niloticus, showed that more than 50 fox genes were expressed in the gonads, and some of these, including foxl2, foxo3, and foxr1 (formerly known as foxn5), were specific to XX females (Yuan et al., 2014). foxl2 and its relatives are known to be key players in ovarian differentiation and oogenesis in vertebrates. Foxl2 is essential for mammalian ovarian maintenance, and it was demonstrated in Nile tilapia, medaka, and zebrafish that foxl2 is also a critical regulator of ovary development and maintenance (Bertho et al., 2016). Further, foxo3 was shown to be required for ovarian follicular development, and its knockout in mice led to sterility in female mutants due to progressive degeneration of the developing oocytes and lack of ovarian reserve of mature oocytes (Hosaka et al., 2004). foxr1 was also found to have sexually dimorphic expression in eels (Anguilla anguilla and Monopterus albus) and marine medaka (Oryzias melastigma), and was predominately observed in the ovaries (Geffroy et al., 2016; Chi et al., 2017; Lai et al., 2015). However, despite these observational studies, the function of foxr1 in vertebrates, especially its role in reproduction, remains unclear. Thus, in this study, we investigated the evolution of foxr1 and its phylogenetic relationship in a wide range of vertebrate species, as well as its biological function using knockout zebrafish models created by the CRISPR/cas9 system to broaden our knowledge of the evolutionary origin of maternal-effect genes and the underlying mechanisms that contribute to reproductive success in vertebrates.

Materials and Methods

Protein databases

Since our model is based on the zebrafish, all gene/protein nomenclatures will be based on those of fish. First, human (Homo sapiens) Fox protein R1 (Foxr1; ENSG00000176302) was used to BLAST for related protein sequences, and the following amino acid data were retrieved from the ENSEMBL database (http://www.ensembl.org/index.html): mouse, Mus musculus; rat, Rattus norvegicus; guinea pig, Cavia porcellus; pig, Sus scrofa; horse, Equus caballus; cow, Bos taurus; panda, Ailuropoda melanoleuca; opossum, Monodelphis domestica; Chinese softshell turtle, Pelodiscus sinensis; armadillo, Dasypus novemcinctus; frog, Xenopus tropicalis (Foxr1 a); fruit fly, Drosophila melanogaster; nematode, Caenorhabditis elegans; sea squirt, Ciona intestinalis; lamprey, Petromyzon marinus; coelacanth, Latimeria chalumnae; spotted gar, Lepisosteus oculatus (Foxr1 a); cod, Gadus morhua (Foxr1 a); fugu, Takifugu rubripes; medaka, Oryzias latipes (Foxr1 a); platyfish, Xiphophorus maculatus; stickleback, Gasterosteus aculeatus; tetraodon, Tetraodon nigroviridis; tilapia, Oreochromis niloticus; zebrafish, Danio rerio (Foxn1 and Foxn3); and cave fish, Astyanax mexicanus. Then, the protein sequence for zebrafish (Foxr1; NP_001096594.1) was found in the NCBI database (http://www.ncbi.nlm.nih.gov). Using this amino acid sequence as bait in BLAST, the bald eagle, Haliaeetus leucocephalu; penguin, Pygoscelis adeliae; crested ibis, Nipponia nippon; swan goose, Anser cygnoides domesticus; American alligator, Alligator mississippiensis; Chinese alligator, Alligator sinensis; python, Python bivittatus; central bearded dragon, Pogona vitticeps; frog, Xenopus laevis and Xenopus tropicalis (Foxr1 b) ; medaka, Oryzias latipes (Foxr1 b); northern pike, Esox lucius (Foxr1 a); rainbow trout, Oncorhynchus mykiss (Foxr1 a); coho salmon, Oncorhynchus kisutch; and Atlantic salmon, Salmo salar, protein sequences were extracted and investigated from the NCBI database. Further, using the protein sequence for zebrafish Foxr1, the following protein sequences were extracted from our previously established PhyloFish online database (http://phylofish.sigenae.org/index.html) (Pasquier et al., 2016) and analyzed along with the others: spotted gar, Lepisosteus oculatus (Foxr1 b); cod, Gadus morhua (Foxr1 b); bowfin, Amia calva; European eel, Anguilla anguilla; butterflyfish, Pantodon buchholzi; sweetfish, Plecoglossus altivelis; allis shad, Alosa alosa; arowana, Osteoglossum bicirrhosum; panga, Pangasius hypophthalmus; northern pike, Esox lucius (Foxr1 b); eastern mudminnow, Umbra pygmae; American whitefish, Coregonus clupeaformis; brook trout, Salvelinus fontinalis (Foxr1 a and b); rainbow trout, Oncorhynchus mykiss (Foxr1 b); European whitefish, Coregonus lavaretus; grayling, Thymallus thymallus; and European perch, Perca fluviatilis. These sequences are compiled in Data S1.

Phylogenetic analysis

The phylogenetic analysis was conducted using the Phylogeny.fr online program with default settings (Dereeper et al., 2008, 2010). Amino acid sequences of 73 Foxr1, Foxr2, Foxn1, and Foxn3 proteins from the above-mentioned species were aligned using the MUSCLE pipeline, alignment refinement was performed with Gblocks, and then the phylogenetic tree was generated using the Maximum Likelihood method (PhyML pipeline) with 100 bootstrap replicates.

Synteny analyses

Synteny maps of the conserved genomic regions of foxr1 and foxr2 were produced with spotted gar as the reference gene using PhyloView on the Genomicus v91.01 website (http://www.genomicus.biologie.ens.fr/genomicus-91.01/cgi-bin/search.pl).

Fish husbandry

Wildtype zebrafish (Danio rerio) of the AB strain were maintained at 25 °C in a central filtration recirculating system with a 12 h light/dark cycle in the INRA LPGP fish facility (Rennes, France). Individual couple pairing was performed by placing a male and a female overnight in a tank with a partition for separation, and in the morning, the divider was removed after which the female released her eggs to be fertilized by the male. All procedures of fish husbandry and sample collection were in accordance with the guidelines set by the French and European regulations on animal welfare. Protocols were approved by the Rennes ethical committee for animal research (CREEA) under approval no. R2012-JB-01.

Quantitative real-time PCR

Tissue samples from two wildtype males and three wildtype females, and 50–200 fertilized eggs at the one-cell stage from 32 wildtype couplings, which were all used as separate biological replicates, were harvested, and total RNA was extracted using Tri-Reagent (Molecular Research Center, Cincinnati, OH, USA) according to the manufacturer’s instructions. The tissue and egg samples were flash-frozen in liquid nitrogen immediately upon harvest and stored at −80 °C until use. Reverse transcription (RT) was performed using one μg of RNA from each sample with the Maxima First Strand cDNA Synthesis kit (Thermo Scientific, Waltham, MA, USA). Briefly, RNA was mixed with the kit reagents, and RT performed at 50 °C for 45 min followed by a 5-min termination step at 85 °C. Control reactions were run without reverse transcriptase and used as negative control in the quantitative real-time PCR (qPCR) study. qPCR experiments were performed with the Fast-SYBR GREEN fluorophore kit (Applied Biosystems, Foster City, CA, USA) as per the manufacturer’s instructions using 200 nM of each primer in order to keep PCR efficiency between 90% and 100%, and an Applied Biosystems StepOne Plus instrument. The StepOne software was used for expression analysis. RT products, including control reactions, were diluted 1/25, and four μL of each sample were used for each PCR. All qPCR experiments were performed in duplicate using technical replicates. The relative abundance of target cDNA was calculated from a standard curve of serially diluted pooled cDNA and normalized to 18S, β-actin, and EF1α transcripts. We considered a Ct variation of around 0.5 as acceptable. The R2 values for foxr1, p21, p27, and rictor were 91.69%, 90.57%, 90.03%, and 90.36%, respectively. Melting curves were used and qPCR products were sequenced for confirmation. The primer sequences can be found in Data S2. The tissue expression of foxr1 was detected using the foxr1 genotyping forward and reverse primers while the mutant form of foxr1 in the CRISPR/cas9-mutated eggs was assessed with the foxr1 qPCR forward and reverse primers.

RNA-seq

Expression profiles in different holostean and teleostean species were obtained using the publicly available PhyloFish database http://phylofish.sigenae.org/index.html. Corresponding RNA-seq data were deposited into Sequence Read Archive of NCBI under accession references SRP044781–SRP044784, SRP045138, SRP045098–SRP045103, and SRP045140–SRP045146. The construction of sequencing libraries, data capture and processing, sequence assembly, mapping, and interpretation of read counts were all performed as previously described (Pasquier et al., 2016). The number of mapped reads was then normalized for the foxr1 gene across the 11 tissues using RPKM normalization.

In situ hybridization

Ovary samples were first fixed in 4% paraformaldehyde overnight, dehydrated by sequential methanol washes, paraffin-embedded, and sectioned to seven μm thickness before being subjected to the protocol. The sections were deparaffinized and incubated with 10 μg/mL of proteinase K for 8 min at room temperature, followed by blocking with the hybridization buffer (50% formamide, 50 μg/mL heparin, 100 μg/mL yeast tRNA, 1% Tween 20, and 5X saline-sodium citrate [SSC]). The probe was diluted to one ng/μL in the hybridization buffer and incubated overnight at 55 °C in a humidification chamber. The probes were synthesized by cloning a fragment of the foxr1 gene into the pCRII vector using the cloning foxr1 forward and reverse primers (Data S2) and Topo TA Cloning kit (Invitrogen, Carlsbad, CA, USA) as per the manufacturer’s protocol. The digoxigenin (DIG)-labeled sense and anti-sense probes were transcribed from Sp6 and T7 transcription sites, respectively, of the vector containing the cloned foxr1 fragment and purified using 2.5M LiCl solution. The purity and integrity of the probes were verified using the Nanodrop spectrophotometer (Thermo Scientific, Waltham, MA, USA) and the Agilent RNA 6000 Nano kit along with the Agilent 2100 bioanalyzer (Santa Clara, CA, USA). The slides were then subjected to two washes each with 50% formamide/2X SSC, 2X SSC, and 0.2X SSC at 55 °C followed by two washes with PBS at room temperature. The sections were subsequently blocked with blocking buffer (2% sheep serum, 3% bovine serum albumin, 0.2% Tween 20, and 0.2% Triton-X in PBS), and the anti-DIG antibody conjugated to alkaline phosphatase (Roche Diagnostics, Mannheim, Germany) was diluted by 1/500 and applied for 1.5 h at room temperature. The sections were washed with PBS and visualized with NBT/BCIP (nitro blue tetrazolium/5-bromo-4-chloro-3-indolyl phosphate).

CRISPR-cas9 genetic knockout

CRISPR/cas9 guide RNA (gRNA) were designed using the ZiFiT (Sander et al., 2010; Mali et al., 2013) online software and were made against two targets within the gene to generate a genomic deletion of approximately 240 base pairs (bp) that spans the last exon which allowed the formation of a non-functional protein. Nucleotide sequences containing the gRNA were ordered, annealed together, and cloned into the DR274 plasmid. In vitro transcription of the gRNA from the T7 initiation site was performed using the Maxiscript T7 kit (Applied Biosystems, Foster City, CA, USA) and of the cas9 mRNA using the mMESSAGE mMACHINE kit (Ambion/Thermo Scientific, Waltham, MA, USA) from the Sp6 site, and their purity and integrity were assessed using the Agilent RNA 6000 Nano Assay kit and 2100 Bioanalyzer. Zebrafish embryos at the one-cell stage were micro-injected with approximately 30–40 pg of each CRISPR/cas9 guide along with purified cas9 mRNA. The embryos were allowed to grow to adulthood, and genotyped using fin clip and PCR that detected the area around the deleted region. The full-length wildtype PCR band was 400 bp, and the mutant band with the CRISPR/cas9-generated deletion was approximately 160 bp using the foxr1 genotyping pair of primers. The PCR bands of the mutants were then sent for sequencing to verify the deletion. Once confirmed, the mutant females were mated with males harboring the vasa::gfp gene, where vasa was fused with gfp at the 3′ end, to produce F1 embryos, whose phenotypes were subsequently recorded. Images were captured with a Nikon AZ100 microscope and DS-Ri1 camera (Tokyo, Japan).

Genotyping by PCR

Fin clips were harvested from animals under anesthesia (0.1% phenoxyethanol) and lysed with 5% chelex containing 100 μg of proteinase K at 55 °C for 2 h and then 99 °C for 10 min. The extracted DNA was subjected to PCR using Jumpstart Taq polymerase (Sigma-Aldrich, St. Louis, MO, USA) and the foxr1 forward and reverse primers that are listed in Data S2.

Statistical analysis

Comparison of two groups was performed using the GraphPad Prism statistical software (La Jolla, CA, USA), and either the Student’s t-test or Mann–Whitney U-test was conducted depending on the normality of the groups based on the Anderson–Darling test. A p-value < 0.05 was considered as significant.

Results

Phylogenetic analysis of Foxr1-related sequences

foxr1 and foxr2 were formerly known as foxn5 and foxn6, respectively (Katoh & Katoh, 2004a, 2004b). To date, there are six reported members of the foxr/foxn family (foxn1, foxn2, foxn3, foxn4, foxr1, and foxr2). To examine the evolution of foxr1, we used a Blast search approach using the zebrafish Foxr1 protein sequence as query in various public databases to retrieve 73 protein sequences from other species that could be related to this protein. All retrieved sequences are compiled in Data S1. Of note, both Foxr1 and Foxr2 protein sequences were retrieved. In order to verify that the retrieved protein sequences (Data S1) were homologous to zebrafish Foxr1, a phylogenetic analysis was performed. Based on the alignment of the retrieved vertebrate Foxr1-related sequences, and using Foxn1 and Foxn3 amino acid sequences as out-groups, a phylogenetic tree was generated (Fig. 1). As shown in Fig. 1, the common ancestor of the vertebrate foxr1 and foxr2 genes diverged from the ancestor of foxn1 and foxn3 genes, and these sequences were clearly observed as two separate clades belonging to actinopterygii (ray-finned fish) and sarcopterygii (lobe-finned fish and tetrapods). In addition, Foxr2 was found only in mammals with no homologs detected in actinopterygii as well as sauropsids and amphibians. Remarkably, despite the wide-ranging presence of Foxr1, no related sequences were observed in invertebrates and chondrichthyans (dogfish and sharks) or in certain species such as chicken (Gallus gallus). On the other hand, several species showed two Foxr1 sequences including the salmonids, rainbow trout (Oncorhynchus mykiss) and brook trout (Salvelinus fontinalis), as well as northern pike (Esox lucius), cod (Gadus morhua), medaka (Oryzias latipes), and spotted gar (Lepisosteus oculatus).

Figure 1 Phylogenetic tree of vertebrate Foxr1 and Foxr2 proteins.

This phylogenetic tree was constructed based on the amino acid sequences of Foxr1 proteins (for the references of each sequence see Data S1) using the Maximum Likelihood method with 100 bootstrap replicates. The number shown at each branch node indicates the bootstrap value (%). The tree was rooted using Foxn1 and Foxn3 sequences. The Foxr1 sequences are in red, Foxr2 sequences are in blue, those of Foxn1 are in green, and Foxn3 sequences are in purple. a and b denotes multiple copies of Foxr1 sequences.

In line with the previous report that stated that foxr2 was absent in tilapia, stickleback, zebrafish, and medaka genomes, we retrieved Foxr2 protein sequences only inmammals using the zebrafish Foxr1 peptide sequence as query. Thus, using zebrafish Foxr1 sequence as the reference protein, we subsequently compared Foxr1 homology with the Foxr1 and Foxr2 sequences from mammals. As shown in Data S3, there was 29–37% identity and 41–53% similarity between all sequences, and there did not appear to be any difference in homology between zebrafish Foxr1 and mammalian Foxr1 and Foxr2 sequences. Further, there was 47–60% identity and 59–77% similarity between mammalian Foxr1 and Foxr2 sequences, indicating that these two proteins are highly similar and probably diverged recently during evolution.

Synteny analysis of foxr1 and foxr2 genes in vertebrates

In order to further understand the origin of the foxr1 and foxr2 genes in vertebrates, we performed a synteny analysis of their neighboring genes in representative vertebrate genomes using the basal actinopterygian, spotted gar, as the reference genome and the Genomicus online database (Fig. 2). We found that between the spotted gar and mammals, there was conserved synteny of the foxr1, upk2, ccdc84, rps25, trappc4, slc37a4, and ccdc153 loci in their genomes. In the frog (Xenopus tropicalis) genome, the foxr1, ccdc153, cbl, mcam, and c1qtnf5 loci were conserved, while in Coelacanth, foxr1, ccdc84, rps25, trappc4, slc37a4, cbl, ccdc153, mcam, c1qtnf5, as well as rnf26 loci were found in the same genomic region as those of the spotted gar. However, amongst the actinopterygians, there was lower conservation of synteny; in zebrafish and cave fish, the foxr1, ccdc84, and mcam loci were conserved while in the other ray-finned fish species, only the foxr1 loci was found. We further analyzed the foxr2 sequences that were found only in mammals, and we demonstrate here that they were all present on the X chromosome with no apparent conserved synteny of neighboring genes to those found in the spotted gar. Our overall analyses suggest that all the foxr-related sequences that were found were homologs, and the foxr gene in fish species probably derived from the ancestral foxr1 gene. Although there was the same degree of protein homology between zebrafish Foxr1 and mammalian Foxr1 and Foxr2 sequences, the phylogenetic tree and synteny analyses showed a clear distinction between them, and the foxr2 gene probably derived from a later single gene duplication or transposon event as previously suggested (Katoh & Katoh, 2004a).

Figure 2 Conserved genomic synteny of foxr1 genes.

Genomic synteny maps comparing the orthologs of foxr1, foxr2, and their neighboring genes, which were named after their human orthologs according to the Human Genome Naming Consortium (HGNC). Orthologs of each gene are shown in the same color, and the chromosomal location is shown next to the species name. foxr1 orthologs are boxed in red while foxr2 orthologs are boxed in blue. Light gray boxes denote the lack of synteny of the corresponding genes in the indicated species as compared to gar.

Expression profiles of foxr1

We next focused our efforts on foxr1 since it has previously been shown in eel, tilapia, and medaka to be gonad specific and thus may have specific functions in reproduction. In order to investigate the potential functions of foxr1, we explored its tissue distribution using two different approaches, qPCR and RNA-seq, the latter of which was obtained from the PhyloFish online database (Pasquier et al., 2016). In zebrafish, we observed from both sets of data that foxr1 mRNA was predominantly expressed in the ovary and unfertilized egg (Figs. 3A and 3B). A high expression of foxr1 transcript was also detected in bone by RNA-seq, but not by qPCR; this could be due to methodological differences as RNA-seq is more sensitive than qPCR since it is designed to detect the sequences all along the transcript while the latter detects just one area of the transcript. Thus, it is possible that a different splice variant of foxr1 exists in bone. By in situ hybridization (ISH), we also demonstrated that foxr1 transcripts were highly expressed in the ovary in practically all stages of oogenesis (Figs. 3C–3E; negative controls, Figs. 3F–3H).

Figure 3 Expression profile of foxr1 in zebrafish.

Tissue expression analysis of foxr1 mRNA in adult zebrafish (A) by quantitative real-time PCR (qPCR) and (B) RNA-seq. Expression level by qPCR is expressed as a normalized value following normalization using 18S, β-actin, and ef1α expression while that by RNA-seq is expressed in read per kilobase per million reads (RPKM). Tissues were harvested from three to four wildtype zebrafish individuals. (C–H) In situ hybridization was performed for foxr1 in zebrafish ovaries from wildtype females. Positive staining is demonstrated using the anti-sense probe against foxr1 (Figs 3C–3E) in blue with 5-bromo-4-chloro-3-indolyl-phosphate/nitro blue tetrazolium as substrate. The negative control was performed with the sense probe (Figs 3F–3H). About 20× magnification; scale bar denotes 90 μm. N = 5 each for foxr1 mutant and control. UF, unfertilized.

Functional analysis of foxr1 in zebrafish

To understand the role of foxr1 during oogenesis and early development, we performed functional analysis by genetic knockout using the CRISPR/cas9 system. One-cell staged embryos were injected with the CRISPR/cas9 guides that targeted foxr1 and allowed to grow to adulthood. Mosaic founder mutant females (F0) were identified by fin clip genotyping and subsequently mated with vasa::gfp males, and embryonic development of the F1 fertilized eggs was recorded. Since the mutagenesis efficiency of the CRISPR/cas9 system was very high, as previously described (Auer et al., 2014; Gagnon et al., 2014), the foxr1 gene was sufficiently knocked-out even in the mutant mosaic F0 females. This was evidenced by the substantially lower transcript level of foxr1 in the F1 embryos as compared to those from control pairings (0.78 ± 0.22 and 1.65 ± 0.04, respectively; Fig. 4A). Thus, the phenotypes of foxr1 (n = 5) mutants could be observed even in the F0 generation. Since neither males nor the females could transmit the mutated genes to future generations (i.e. all the surviving embryos were WT), all of our observations were obtained from the F1 generation, which were the fertilized eggs derived from the mosaic female foxr1 mutants.

Figure 4 CRISPR/cas9 knockout of foxr1 in zebrafish.

(A) Normalized expression level of foxr1 transcript by quantitative real-time PCR (qPCR) in the fertilized zebrafish eggs from crosses between foxr1 mutant females and vasa::gfp males. (B) Developmental success (% survival) at 24 h post-fertilization (hpf) as measured by the proportion of fertilized eggs that underwent normal cell division and reached normal developmental milestones based on Kimmel et al. (1995) from crosses between foxr1 mutant females and vasa::gfp males. (C) Frequency of foxr1 mutant phenotypes in the F1 eggs between crosses of foxr1 mutant females and vasa::gfp males. #Embryos did not develop at all (please refer to Figs 5E–5H). +Embryos had a partially cellularized blastodisc that was sitting atop an enlarged syncytium (please refer to Figs 5I–5L). The graphs demonstrate representative data from a single clutch from a mutant female. qPCR data were normalized to 18S, β-actin, and ef1α. N = 5 each for foxr1 mutant and control. All assessments were performed from at least three clutches from each mutant. ** p < 0.01 by Mann–Whitney U-test. Control = eggs from crosses of wildtype females with vasa::gfp males; foxr1 = eggs from crosses of foxr1 mutant females with vasa::gfp males.

We observed that most of the embryos from the foxr1 mutant females had a very low developmental success at 24 hpf (23.2 ± 8.4% vs. 85.2 ± 9.2% in controls; p < 0.008) (Fig. 4B). The frequency of the mutation in the mutant females is demonstrated in Fig. 4C, and it was observed that three of the mutants produced abundant non-developing eggs that remained non-cellularized, reflecting their failure to undergo cell division (Figs. 5E–5H). The eggs derived from these three foxr1 mutant females (Fig. 4: foxr1-1, foxr1-2, and fox1-3) did not undergo any cell division at two hpf and continued to display a complete lack of development up to eight hpf. By 24 hpf, these non-developing eggs that failed to divide were all dead. In addition, two of the mutants (Fig. 4: foxr1-4 and fox1-5) produced developmentally incompetent eggs with two phenotypes; those with a non-cellularized morphology (Figs. 5E–5H), and another population that developed albeit with an abnormal morphology (Figs. 5I–5L). These fertilized and developing embryos were structurally abnormal, with unsmooth and irregularly-shaped yolk as well as asymmetrical cell division that culminated into a blastodisc with a group of cells on top of an enlarged syncytium (Fig. 5K, arrow). These eggs underwent developmental arrest at around four hpf or the MBT and appeared to regress with further expansion of the syncytium (Figs. 5J–5K) until death by 24 hpf.

Figure 5 Effect of foxr1 deficiency on zebrafish embryogenesis.

Representative images demonstrating development of fertilized eggs from crosses between control (A–D) and foxr1 (E–L) females and vasa::gfp males from 2 to 24 h post-fertilization (hpf). In the control eggs, the embryos were at 64-cell (A), oblong (B), germ ring (C), and 24-somite (D) stages according to Kimmel et al. (1995). Eggs from foxr1 mutant females were non-developing with a non-cellularized morphology (E–H) or developing with an abnormal morphology (I–L). (A, E, I) = images taken at two hpf; (B, F, J) = images taken at four hpf; (C, G, K) = images taken at six hpf; (D, H, L) = images taken at 24 hpf. Scale bar denotes 500 μm. The arrow demonstrates an abnormally cellularized blastodisc that was sitting atop an enlarged syncytium. (M) Genotypic analysis of the eggs from crosses of foxr1 mutant females and vasa::gfp males to determine fertilization status. The gfp and vasa primers produced a band that was 1,333 base pairs in size. Detection of the npm2b gene (band size = 850 base pairs) was used as a control. Con = eggs from crosses of wildtype females with vasa::gfp males; foxr1 = eggs from crosses of foxr1 mutant females with vasa::gfp males. N = 5 each for foxr1 mutant and control.

The observed phenotype of the foxr1 mutant-derived uncellularized eggs was very similar to previously described unfertilized eggs (Dekens et al., 2003). Thus, the foxR1 mutant females were mated with vasa::gfp males, and the genotype of their progeny was assessed for the presence of the gfp gene, which would only be transmitted from the father since the mutant females did not carry this gene. We found that these uncellularized eggs from the foxr1 mutant females did indeed carry the gfp gene (Fig. 5M), which indicated that some or all of them were fertilized, but were arrested from the earliest stage of development and did not undergo any cell division. These novel findings showed for the first time that foxr1 is essential for the developmental competence of zebrafish eggs, and is therefore a crucial maternal-effect gene.

In order to delve into the possible mechanisms that may be involved in the reduced reproductive success of the foxr1 mutants, we investigated the expression levels of p21, p27, and rictor, which were previously reported to be repressed by the Foxr1 transcription factor in mice (Santo et al., 2012). We found that there was substantially increased expression of p21 (4.82 ± 1.09 vs. 0.25 ± 0.04 in controls; p < 0.002) while that of rictor was significantly decreased (0.87 ± 0.13 vs. 1.60 ± 0.25 in controls; p < 0.01) in the foxr1 mutant-derived eggs as compared to eggs produced by wildtype females (Figs. 6A–6C). The expression of p27 was unchanged between the two groups. These results were in line with the growth arrested phenotype that was observed in the uncellularized and developmentally challenged eggs from the foxr1 mutant females.

Figure 6 Expression profiles of (A) p21, (B) p27, and (C) rictor in eggs from foxr1 mutant females.

Fertilized eggs from foxr1 mutant females were subjected to qPCR for examination of the transcript levels of p21, p27, and rictor. The graphs demonstrate representative data from a single clutch from a mutant female. Data were normalized to 18S, β-actin, and ef1α. N = 4 each for foxr1 mutant and control, at least two clutches were used from each animal, and each experiment was performed in duplicate. *p < 0.05, **p < 0.01 by Mann–Whitney U-test.

Discussion

In this study, we first investigated the evolutionary history of foxr1 in order to gain perspective into its phylogenetic relationship among homologs from a wide range of species and to clarify its origins. Using the zebrafish protein sequence as query to search for homologs in other species, we retrieved Foxr1 sequences from a broad variety of vertebrates, including actinopterygii, sarcopterygii, and sauropsids which suggested the importance of this protein in most vertebrates. We also retrieved Foxr2 sequences from many vertebrates due to its high similarity to the zebrafish Foxr1 peptide (Data S3), although we and others demonstrated that the foxr2 gene is absent from all actinopterygii and sauropsid species, and can only be found in mammals. Evidence from the phylogenetic analyses showed a clear distinction in derivation of the actinopterygian foxr1 and the mammalian foxr2; the divergence of the ancestral foxr1 gene in actinopterygii from that of the sarcopterygii and sauropsids occurred quite early in evolution, while the divergence of mammalian foxr1 and foxr2 is a much more recent event (Fig. 1). Further, the synteny analysis (Fig. 2) showed that there was much conservation of genomic synteny surrounding the foxr1 loci between the basal actinopterygian, spotted gar, and actinopterygii and sauropsids, while the neighboring loci around the foxr2 were completely different in comparison to those next to foxr1 which suggested that foxr2 originated from a recent gene duplication or transposition event as previously proposed (Katoh & Katoh, 2004a). We also found that in a small subset of species [rainbow trout (Oncorhynchus mykiss) and brook trout (Salvelinus fontinalis), as well as northern pike (Esox lucius), cod (Gadus morhua), medaka (Oryzias latipes), and spotted gar (Lepisosteus oculatus)], two Foxr1 sequences were observed. This suggested that independent gene duplication events occurred in these lineages. It is also possible that foxr1 was duplicated in the ancestral actinopterygii followed by the loss of one copy in a lineage-dependent manner. Finally, it appeared that the different whole-genome duplication events, the teleost-specific genome duplication and salmonid-specific genome duplication did not impact the current foxr1 gene diversity because in most species, only one foxr1 gene was retained. The presence of two foxr1 sequences in the above-mentioned species could also be due to independent and phylum-specific gene retention or independent gene duplication events that occurred only in these species or technical differences due to different sequencing procedures. Further phylogenetic, synteny, and functional analyses on the two copies of foxr1 in these species are warranted in order to verify the functionality of both genes.

The essentialness of foxr1 was suggested by the wide-ranging presence of this gene in most vertebrates and the retention of a single copy in most teleosts despite multiple whole genome duplication events, but its biological function is still largely unknown. Previous reports have demonstrated the predominant expression of foxr1 mRNA in the ovary of medaka, eel, and tilapia (Yuan et al., 2014; Geffroy et al., 2016; Lai et al., 2015), but it was found mostly in the male germ cells and spermatids in mouse and human (Petit et al., 2015). It was further shown to be abundantly expressed in the early cleavage and gastrula stages of Xenopus embryos, but absent in post-gastrula stages due to rapid degradation of its mRNA, indicating that it is a maternally-inherited transcript (Schuff et al., 2006). Thus, the foxr1 gene may play different roles in reproduction in teleost fish/amphibians and mammals, suggesting that foxr2 in mammals may have evolved to have comparable functions to the teleost/amphibian foxr1 while mammalian foxr1 is mostly involved in male reproduction and development. Of note, the mammalian foxr2 is thus far observed only on the X chromosome highly indicating a function in female reproduction (Fig. 2). Future studies to test this are necessary to confirm the function of foxr2. To confirm these results found in other teleosts in zebrafish, we first examined the expression profile of foxr1 in various tissues, and we showed by qPCR as well as by RNA-seq that there was also an ovarian-specific expression of foxr1 and negligible amount in the testis as in the other fish species. By ISH, we found that the foxr1 transcript was progressively stored in the growing oocytes from the very early stages (Figs. 3C–3D, arrows) to later staged oocytes (Figs. 3D–3E), and could be found abundantly in mature fertilized eggs (Figs. 3B and 4A). These results demonstrated that foxr1 is one of the maternal products that is deposited into the developing oocytes during oogenesis in zebrafish as observed in other fish species. These findings suggest that foxr1 may function as a maternal-effect gene such as npm2a, npm2b (Cheung et al., 2017), bucky ball (Bontems et al., 2009), futile cycle (Lindeman & Pelegri, 2012), and wnt (Nicol & Guiguen, 2011).

In recent years, there has been an increasing identification of maternal-effect genes and their functions, but there is still limited information on the role of genes that regulate early embryogenesis. Previous studies have found that irreducible, indivisible, atomos, cellular island, cellular atoll, and nebel genes among others have roles in early embryonic cell cleavage and cell division, but their protein identity as well as their regulatory mechanisms are largely unknown or not yet clarified (Dosch et al., 2004; Pelegri et al., 1999). Having established that foxr1 was indeed a maternal factor, we investigated its function via mutagenic analysis with CRISPR/cas9. We used the F0 mosaic mutant females that were shown to have a decreased level of foxr1 mRNA for analysis due to the difficulty in transmitting the mutated foxr1 gene to future generations as both the F0 foxr1 mutant females and males produced mostly non-viable progeny, and the surviving descendants were all of wildtype genotype. We found that the foxr1 mutant females produced bad quality eggs, and the developmental success of their progeny was very low, similar to that of foxl2 and foxo3 mutants. These latter two genes have been previously shown to be crucial in ovarian determination since they are necessary for the development of the ovary as well as ovarian fate maintenance via suppression of male cues (Bertho et al., 2016; Hosaka et al., 2004). Deficiency in either of these two genes results in small ovaries and disorganized follicles that cannot mature as well as the appearance of testis-specific cells. Thus, it is possible that foxr1 may be also required for proper ovarian development and function, as also observed for foxl2 and foxo3 mutants, and further histological analysis on ovaries from the foxr1 mutants is warranted. In addition, we found that the foxr1 mutant-derived eggs were non-cellularized and did not undergo subsequent cell division despite being fertilized, as also observed for eggs derived from npm2b mutant females (Cheung et al., 2017). This suggested that their defect did not lie in the capability to be fertilized, as seen in slc29a1a and otulina mutants (Cheung et al., 2018), but in the cell cycle and proliferation processes. Thus, we investigated the expression profiles of p21, p27, and rictor, which are all cell cycle and cell survival regulators, since Santo et al. (2012) had previously knocked down foxr1 using short hairpin RNAs in mammalian cells and found it to be a transcriptional repressor of them. In this report, we also observed a dramatic increase in p21 transcript in the eggs from foxr1 mutant females, although the expression of p27 was unchanged, while that of rictor was decreased. Both p21 and p27 are well known cell cycle inhibitors, and rictor is a component of the mTOR (mammalian target of rapamycin) complex that is a major regulator of cell growth and proliferation (Tateishi et al., 2012; Peponi et al., 2006). In fact, mitogens or some survival signal activates a survival cascade, such as the PI3K/Akt pathway, which is activated by the rictor-mTOR complex and promotes cell growth through repression of the negative cell cycle modulators, including p21 and p27 (Sarbassov, Ali & Sabatini, 2005). Thus, our findings were in line with a phenotype of growth arrest and anti-proliferative effects as seen in our eggs derived from foxr1 mutant females. The different results that we observed as compared to those from Santo et al. were probably due to species- and cell type-specific effects.

In this study, we showed that foxr1 is found in a wide-range of vertebrates and is homologous to the foxr1 genes found in other species. In teleosts, foxr1 expression is found predominately in the ovary while in mammals, it appears to be specific to the male germline (Petit et al., 2015). We also found that foxr1 is a novel maternal-effect gene and is highly expressed in the developing oocytes as well as accumulated in mature eggs to be used in early embryogenesis. Our findings suggest that maternally-inherited foxr1 may be required for the first few cleavages after fertilization for proper cell growth and proliferation possibly via p21 and rictor, since deficiency in foxr1 leads to either complete lack of or abnormal cell division culminating to early death in the fertilized egg. Further molecular analyses to disrupt p21 and rictor expression in foxr1 mutants to investigate their function in these mutants are necessary. Thus, the results of this study are the first to establish a link between egg quality, the control of early cell cycle, and the molecular regulatory mechanisms via the potential transcriptional regulator, FoxR1.

Conclusions

Our study shows for the first time that foxr1 is an essential maternal-effect gene and is required for proper cell division and survival possibly via the p21 and mTOR pathways. These novel findings will broaden our knowledge on the functions of specific maternal factors stored in the developing egg and the underlying mechanisms that contribute to reproductive fitness.

Supplemental Information

Supplemental Information 1 Foxr1-related protein sequences used in the phylogenetic analysis.

Species, database, accession number and amino acid sequence are shown.

Click here for additional data file.

Supplemental Information 2 qPCR and PCR primer sequences.

For each gene, primer sequence is provided.

Click here for additional data file.

Supplemental Information 3 Percentage homology between Foxr1 and Foxr2 protein sequences.

Sequence positivity and similiarity is shown for the proteins of the different species.

Click here for additional data file.

Supplemental Information 4 Raw data for Figs. 3–6.

Click here for additional data file.

Supplemental Information 5 Uncropped blot Figure 5.

Click here for additional data file.

The authors would like to thank all the members of the LPGP laboratory for their assistance. We thank JJ Lareyre for the use of the vasa::gfp males for the fertilization assays.

Additional Information and Declarations

Competing Interests

Author Contributions

Animal Ethics

Data Availability

The authors declare that they have no competing interests.

Caroline T. Cheung conceived and designed the experiments, performed the experiments, analyzed the data, prepared figures and/or tables, authored or reviewed drafts of the paper, approved the final draft.

Amélie Patinote performed the experiments, approved the final draft.

Yann Guiguen conceived and designed the experiments, approved the final draft.

Julien Bobe conceived and designed the experiments, analyzed the data, authored or reviewed drafts of the paper, approved the final draft.

The following information was supplied relating to ethical approvals (i.e., approving body and any reference numbers):

Protocols were approved by the Rennes ethical committee for animal research (CREEA) under approval no. R2012-JB-01.

The following information was supplied regarding data availability:

The raw data are provided in the Supplemental Files.

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
