# Peer review of "foxr1 is a novel maternal-effect gene in fish that is required for early embryonic success"

_PeerJ, doi:10.7717/peerj.5534_

## Round 0.1 · original submission · Major Revisions

Thank you for submitting your manuscript to PeerJ. Based on the three accompanying reviews and my own reading I invite you to resubmit after making major revisions. All reviewers were positive about the contributions of this work and provide comments that I think would improve the manuscript when addressed. Reviewer one has the most serious concerns related to confirmation of CRISPR knockout and presentation of the qPCR data, as well as several other points that should be addressed. Reviewer 3 raises concerns about your conclusion that foxr1 affects embryogenesis through its regulation of p21 and rictor. They also note that your discussion as written primarily restates the results, and needs to be re-worked.

Please address all reviewer comments in your resubmission. Please also consider these additional points:

Consider adding some additional clarification to the qRT-PCR methods:
- How were samples handled/stored prior to purification of RNA?
- Were the duplicates mentioned technical replicates or biological replicates? I assume these are technical replicates as you mention collecting RNA from two males and 3 females, but this could be clarified. You also mention 32 couplings? How many eggs were used per coupling? Were eggs from all 32 couplings used as separate biological replicates? This point can be clarified.
- What amount of variation in Ct between technical duplicates was acceptable? What proportion, if any, had higher variation?
- It would be helpful to report qPCR primer efficiencies and R2 values. What were the sizes of each product? Where melt curves used and products sequenced to confirm their identity? Could you provide the accession numbers for each target gene?
- What algorithm/software was used to calculate relative abundance?

Since figure 3 uses previously cited RNA-Seq data it may be worth specifically mentioning this in the methods when reference 14 is cited. As written, citation 14 only seems to refer to the method, and not the fact that these data have been previously described, in part, in the literature.

Line 66: It is unclear what “which” is referring to.

Line 67-68: the statement “especially its role in reproduction” needs commas on either side.

Line 70: delete “in order” and change to “broaden our knowledge of . . . “

Line 92: Do you mean the same species’ protein sequences were also extracted from NCBI as from ENSEMBLE? This could be clarified?

Line 104: Could you define what custom setting were use in Phylogeny.fr for each step, or whether all custom settings were used?

Reviewer 1 ·

Basic reporting

The underlying claim that foxr1 is a evolutionarily conserved gene necessary for early embryonic development seems sound given the multiple mutant offspring analysis. The results relate to this hypothesis, but there is need for some work on the description of experimentation (see section below) before a reader can be convinced that care in experimentation has been taken to warrant all the claims the authors make about mechanism. Additionally speculative parts of the paper need to be addressed.

The UF in Figure three should be described as unfertilized in the figure legend for clarity.

Experimental design

The recording of how the protein sequence for the different foxr genes were obtained should be clarified. The statement “amino acid data were extracted” in the methods section is insufficient to help the reader understand how the databases were queried (BLAST stringencies should be described or method cited). This will be helpful for someone interpreting the lack of Foxr1 in many species (lines 211-214) and to reproduce the results.

The verification of Cas9 knockout is insufficiently documented, given the tendency of smaller PCR templates to be amplified when in the same reaction as larger templates (in this case, mutant and wild-type respectively.

Description of the qPCR experiments in Figures 4 and 6 line 267-268 and 299-304 should be presented with scatter plots to help the reader understand the distribution of the data and a statement of which mutant(s) produced the experimental embryos. This is especially important given the difference in developmental potential through the blastula stages of two developmental types of presumptive heterozygotes, clutches could have been divided into groups of non-cellularized, cellularized with developmental defects, and normally developing embryos.

Description of male transmission of mutated foxr1 genes is lacking or confusing (lines 271 and 364-365). How many F0 males were mated? What were the results? Why no qPCR data on testes, given 2 males were dissected for tissue analysis?

Line 295 should read Fig 5M rather than 4M.

The gels shown in Figure 5 and in peerj-27280-Uncropped_image_Figure_5 do not appear to be the same exposure of the same gel. The bands look dissimilar.

Validity of the findings

The Cas9 deletion approach and the ability of the authors to detect fertilization by assaying a transgene only from sperm is a compelling assay for fertilization in non-developing embryos and supports the conclusion of a maternal effect for foxr1.

I am not expertly qualified to comment on the phylogenetic analysis of this paper, but the tree seems like a reasonable fit to the data. Would a second BLAST search for homologs using another species than zebrafish be warranted to validate the results found searching only with zebrafish proteins?

I think the authors meant the F1 generation at line 272 since the F0 generation were viable and fertile (excepting foxr1-2) demonstrating no direct phenotype assayed by the authors.

A speculation at lines 217-218 and again at 330-333 is that the different sequences found in fish comes from a gene duplication event is not supported by the data. This is troublesome given that syntenic analysis is obviously done on one of the Foxr1 protein sequences from gar and medaka etc., but not the other. Given the apparent ability of the authors to compare syntenic regions, an explanation about the other sequences and their synteny is warranted. Only sequence obtained from different databases resulted in unique isoforms of the sequence. When speculating, the authors should provide more than one interpretation, and where possible how one would test between the possibilities (lines 340-342 makes the vague statement “Further analyses on the two copies” without explaining the analyses).

The word penetrance (line 276) is used incorrectly in a genetics paper as penetrance is the condition when an phenotype-linked allele is present, but the affected organism does not exhibit a phenotype. The authors had made the point in the paragraph preceding that all surviving embryos were WT. This indicates that the F0 generation were mosaic and many eggs did not have foxr1 disruption. I suggest the word frequency instead.

A comment on the differences seen between qPCR levels and RNA-seq levels in bone is warranted.

The sentence of lines 387-388 is confusing with verb subject disagreement.

The expression claim in lines 388-389 needs citation.

The conclusion statement on 393 and repeated at 399 “via the p21 and mTOR pathways” has been correlated, but not tested in this paper.

Opossum Foxr2 seems given the wrong color in Figure 1. Should it be blue like Armadillo?

Reviewer 2 ·

Basic reporting

This is a well-designed and described study from a prominent fish reproduction biologist.
The major findings included:
The presence of foxr1 in a number of species.
In zebrafish, foxr1 has predominately ovarian specific expression.
In zebrafish oocytes, foxr1 is expressed early and accumulates through oogenesis.
When foxr1 is knocked out, oocyte are still fertilized but have problems undergoing normal cell division. This is due to an increase in p21 and a decrease in rictor.

I thought this was a well written paper with a good depth to the introduction and very straightforward study design and results:
Minor issues
A few awkward or confusing sentences:
*note: my complete review download has different line numbers than the individual paper download.

Line 43: In vertebrates, maternal products including transcripts, proteins, and other biomolecules are necessary for kick-starting early embryonic development until the mid-blastula transition (MBT) when the zygotic genome is activated [1].

Line58: foxl2 and its relatives are known to be key players in ovarian differentiation and oogenesis in vertebrates; it is essential for mammalian ovarian maintenance and through knockout experiments, it was demonstrated that foxl2is a critical regulator of sex determination by regulating ovary development and maintenance also in Nile tilapia, medaka, and zebrafish[9]

Line 333: It is also possible that foxr1 was duplicated in the ancestral actinopterygii and subsequent gene losses in bowfin as well as in the teleosts especially following the multiple gene duplication events such as the teleost-specific whole genome duplication (TGD) and salmonid-specific whole genome duplication (SSGD).

Line 345: Previous reports have demonstrated the predominant expression of foxr1 mRNA in the ovary of medaka, eel, and tilapia[8,11,13], but in the male germ cells and spermatids in mouse and human[24].

Vasa::gfp line wasn’t described when it was first mentioned, but was later in the text. Helpful to know why this was used earlier for people outside of the field of maternal-effect genes

Problem with citation 4
Years missing in citations 9 and 11

Experimental design

Well designed. I have a few minor comments:
No section in the methods on animal husbandry.
npm2b mutants are mentioned on Line 288 but without any background as to what or why they were mentioned. This extends to Fig. 5M
Tissue samples came from 2 males and 3 females. It is not clear why only 2 males were used. Aren’t you treating them as separate groups since you are looking at a sex specific expected phenotype? Furthermore, the way they are reported in Fig. 3, it appears that they are just grouped together. Clarity in the description in the methods would probably clarify the design better and make more clear Fig. 3. Furthermore in the legend for Fig. 3, it says that tissues were harvested from 3 to 4 WT zebrafish?
Fig. 4C: there are no WT controls to compare this information to. You seems to have this information for 4B, therefore it should be added to Fig.4C for comparison.
Fig. 4C the cross indicates that all zebrafish in these groups showed the exact same phenotype of a blastodisc sitting atop an enlarged syncytium? Or is this just an example of a developmental defect. Clarity would be good for this.
Fig. 5M. Why was npm2b used as a control. This isn’t clear since it is also missing from the paper.

Validity of the findings

I am concerned about the uncropped blot. I can’t tell what I am looking at in the cropped version (compared to the uncropped blot). I think just a label on the uncropped blot would allow reassurance.

·

Basic reporting

The manuscript “foxr1 is a novel maternal-effect gene in fish that regulates
embryonic cell growth via p21 and rictor” by Cheung et al. identifies a function for the transcription factor foxr1 in oogenesis and early embryogenesis. They demonstrate that foxr1 mRNA is provided maternally, consistent with this role. Further, through loss-of-function studies using Crispr-Cas9, they find that the number of normal embryos produced from a foxr1 Crispr-Cas9 edited female and a male WT at the foxr1 locus (carrying the vasa::gfp transgene, which should be expressed in the fertilized F1 progeny) is significantly reduced. Phenotypes of the progeny include failure to undergo any cleavages and abnormal cleavages, both of which result in death of the embryos by 24 hours post spawning. As expected from a mosaic parent, a proportion of the embryos are normal. Overall, this is a well done study that will be of interest to the scientific community, and especially to those interested in maternal effect genes and early development. However, there are some changes needed before the manuscript is ready for publication.

Experimental design

The experimental designs are strong. However, in a few places, additional information is needed.
A. How many biological replicates were done for the RNA-seq experiments?
B. For the GFP PCR on F1 progeny with no cell divisions (Figure 5), were the PCRs done on single embryos or groups? This may affect the interpretation of the results.

Validity of the findings

Major concerns.
1. The claim that Foxr1 functions through regulation of p21 and rictor is not supported by the data in this manuscript. The authors demonstrate that expression of these two genes is altered in the F1 progeny of the foxr1 Crispr-Cas9 edited female X vasa::gfp male. There is no functional data to demonstrate that these changes in gene expression are causing the observed phenotypes in the progeny. The authors should change their title to focus only on their research on Foxr1, they should take the p21 and rictor reference out of the title. There also several places in the Discussion that need to be revised to make it clear that Foxr1 acting through p21 and rictor is a hypothesis or working model only. These places are marked in the manuscript pdf.

2. The Results and Discussion sections are somewhat redundant with each other. The authors should follow the conventions of these sections and make sure text is in the appropriate section.

Minor concerns
1. In several places, the figure legends are not complete or need clarification. These places are marked on the pdf of the manuscript.

2. Similarly, there are several places in the text that need to be revised to improved clarity. These places are also marked in the pdf of the manuscript.

---

## Round 0.2 · accepted · Accept

Thank you for your careful consideration of all the reviewers’ comments and your work to revise the manuscript. I am happy to now accept your manuscript for publication in PeerJ. Congratulations on the great work.

Please consider the few edit suggestions and questions left by Reviewer 3 as you prepare your final draft while in Production.

Thank you for addressing all of my comments about your methods. Please consider adding the following details to the final text:
1. A mention that melt curves were used and qPCR products sequenced for confirmation
2. Stating that the StepOne software was used for expression analysis
3. That default settings were used in Phylogeny.fr

You will be given the option to make the reviews of your manuscript available to readers. Please consider doing so as this review record can be a great resource for readers of your paper and contributes to more transparent science.

Thank you again for your contribution.

# Reviewer 1 ·

Basic reporting

no comment

Experimental design

no comment

Validity of the findings

no comment

Additional comments

The underlying claim that foxr1 is a evolutionarily conserved gene necessary for early embryonic development seems sound given the multiple mutant offspring analysis. Care has been taken to separate speculation from supported claims. BLAST analysis appears sound. Description of Cas9-mediated gene knockout is described sufficiently. The data plots are easily readable and typographical errors have been corrected. F0 generation fish were viable giving rise to fertilizable eggs without ability to develop. Gene deletion was confirmed by PCR and fertilization by transgene detection in non-developing embryos and supports the conclusion of a maternal effect for foxr1. The authors have rightly backed off of any causation claims involving rictor and p21. This manuscript is acceptable for publication.

Reviewer 2 ·

Basic reporting

.

Experimental design

.

Validity of the findings

.

Additional comments

I am fine with the changes made to the paper and think it is ready to go.

·

Basic reporting

The manuscript is now acceptable for publication, with the exception of a few sentences that still need to be revised:
Questions:
1. Figure legend 1:
"The Foxr1 sequences are in red, Foxr2 sequences are in blue, those of Foxn1 are in green, and Foxn3 sequences are in purple. a and b denote multiple copies of Foxr1 sequences in fish."
-a and be are also included for Xenopus tropicalis-does this mean the same as for the fish?.

2. Section 310
"In addition, two of the mutants produced developmentally incompetent eggs with two phenotypes; those with a non-cellularized morphology (Fig 5E-H), and another population that developed albeit with an abnormal morphology (Fig 5I-L). "
Are these two mutants included in the three mutants that are discussed earlier in this paragraph, or are these different fish?

Suggested revisions
Section 235
Remarkably, despite the wide-ranging presence of Foxr1, no related sequences were observed in invertebrates and chondrichthyans (dogfish and sharks) or in certain vertebrate species such as chicken (Gallus gallus).

Section 300
Since neither males nor the females could transmit the mutated genes to future generations (ie. all the surviving embryos were WT), all of our observations were obtained from the F1 generation, which were the fertilized eggs derived from the mosaic female foxr1 mutants.

Section 305
The frequency of the mutation in the mosaic females is demonstrated in Fig 4C, and it was observed that 3 of the mosaic fish produced abundant non-developing eggs that remained non-cellularized, reflecting their failure to undergo cell division, suggesting that they had germ-line transmission of the mutation (Fig 5E-H). The eggs derived from these 3 foxr1 mutant females did not undergo...

Section 315
These embryos underwent developmental arrest at around 4 hpf or the MBT and appeared to regress with further expansion of the syncytium (Fig 5J-K) until death by 24 hpf.

Section 340
These results were in line with the growth arrested phenotype that was observed in the uncellularized and developmentally challenged eggs from the foxr1 mutant females.

Experimental design

All of my concerns have been addressed.

Validity of the findings

All of my concerns have been addressed.

Additional comments

All of my concerns have been addressed in the revised manuscript.

---

## Author Rebuttal · Round 0.2

# Editor's Comments

Thank you for submitting your manuscript to PeerJ. Based on the three accompanying reviews and my own reading I invite you to resubmit after making major revisions. All reviewers were positive about the contributions of this work and provide comments that I think would improve the manuscript when addressed. Reviewer one has the most serious concerns related to confirmation of CRISPR knockout and presentation of the qPCR data, as well as several other points that should be addressed. Reviewer 3 raises concerns about your conclusion that foxr1 affects embryogenesis through its regulation of p21 and rictor. They also note that your discussion as written primarily restates the results, and needs to be re-worked.

Please address all reviewer comments in your resubmission. Please also consider these additional points:

Consider adding some additional clarification to the qRT-PCR methods:
- How were samples handled/stored prior to purification of RNA? The tissue and egg samples were flash-frozen in liquid nitrogen immediately upon harvest and stored at -80$^{\circ}$C until use. This is now indicated in the revised manuscript.
- Were the duplicates mentioned technical replicates or biological replicates? I assume these are technical replicates as you mention collecting RNA from two males and 3 females, but this could be clarified. You also mention 32 couplings? How many eggs were used per coupling? Were eggs from all 32 couplings used as separate biological replicates? This point can be clarified. The duplicates were biological replicates. This is now indicated in the revised manuscript.
- What amount of variation in Ct between technical duplicates was acceptable? What proportion, if any, had higher variation? We considered a Ct variation of around 0.5 as acceptable. We have added this statement to the revised manuscript. The qPCR experiments were performed at least 2 times, and every time, the results were the same. We showed the results from the experiments with the lowest variation. We had a low proportion of samples with high variation, less than 10%.
- It would be helpful to report qPCR primer efficiencies and R2 values.
We have added the following data to the revised manuscript.
foxr1 (Fig3A & 4A) : $R^2$ = 91.69%
p21 (Fig6A) : $R^2$ = 90.57%
p27 (Fig6B) : $R^2$ = 90.03%
rictor (Fig6C) : $R^2$ = 90.36%
What were the sizes of each product?
We have added the following data to Supplementary Data 2 in the revised manuscript.
foxr1  (Fig3A & 4A): 57 bp for WT, no amplification for mutant
p21 (Fig6A) : 198 bp
p27 (Fig6B): 139 bp
rictor (Fig6C) : 197 bp
Where melt curves used and products sequenced to confirm their identity?
Yes melting curves were used to confirm that only one product was made and the products were sequenced to confirm their identity as mentioned in the paper.
 Could you provide the accession numbers for each target gene?
We have added the following data to Supplementary Data 2 in the revised manuscript.
foxr1  (Fig3A & 4A): ENSDARG00000004864

p21 (Fig6A) : ENSDARG00000076554
p27 (Fig6B): ENSDARG00000054271
rictor (Fig6C) : ENSDARG00000002020
- What algorithm/software was used to calculate relative abundance? We used standard curves to calculate relative abundance using the internal software that came with the Applied Biosystems StepOne Plus instrument. For each gene in each experiment, a standard curve was made with 5 1:2 dilutions and water control.

Since figure 3 uses previously cited RNA-Seq data it may be worth specifically mentioning this in the methods when reference 14 is cited. As written, citation 14 only seems to refer to the method, and not the fact that these data have been previously described, in part, in the literature. The RNA-seq data in Fig.3 have not been previously described, we newly demonstrated here the data on foxr1. We datamined the publicly available RNA-seq data for the foxr1 data.

Line 66: It is unclear what "which" is referring to. This has been modified; the "which" has been replaced with "and" to clarify our meaning.

Line 67-68: the statement "especially its role in reproduction" needs commas on either side. This has been corrected.

Line 70: delete "in order" and change to "broaden our knowledge of . . . " This has been modified.

Line 92: Do you mean the same species' protein sequences were also extracted from NCBI as from ENSEMBLE? This could be clarified? Multiple sequences were identified from the different databases so they are now distinguished by being labeled as "a" and "b" in Fig 2.

Line 104: Could you define what custom setting were use in Phylogeny.fr for each step, or whether all custom settings were used? No custom settings were used except for 100 bootstrap.

# Reviewer 1 (Anonymous)

## Basic reporting

The underlying claim that foxr1 is a evolutionarily conserved gene necessary for early embryonic development seems sound given the multiple mutant offspring analysis. The results relate to this hypothesis, but there is need for some work on the description of experimentation (see section below) before a reader can be convinced that care in experimentation has been taken to warrant all the claims the authors make about mechanism. Additionally speculative parts of the paper need to be addressed.

The UF in Figure three should be described as unfertilized in the figure legend for clarity. This has been added.

## Experimental design

The recording of how the protein sequence for the different foxr genes were obtained should be clarified. The statement "amino acid data were extracted" in the methods section is insufficient to help the reader understand how the databases were queried (BLAST stringencies should be described or method cited). This will be helpful for someone interpreting the lack of Foxr1 in many species (lines 211-214) and to reproduce the results. New descriptions have been added.

The verification of Cas9 knockout is insufficiently documented, given the tendency of smaller PCR templates to be amplified when in the same reaction as larger templates (in this case, mutant and wild-type respectively. We are not sure we understand this comment. The presence of mutation in the genome was checked using PCR. We agree with this reviewer than small size PCR templates are likely to be preferentially amplified. In our case, this was favorable and allowed the detection of mutations in injected fish. In contrast, the wild-type controls lacked the small-size (160 bp) band present in mutants and only displayed the normal size band (400 bp). We would be happy to provide corresponding gel pictures if necessary..

Description of the qPCR experiments in Figures 4 and 6 line 267-268 and 299-304 should be presented with scatter plots to help the reader understand the distribution of the data and a statement of which mutant(s) produced the experimental embryos. This is especially important given the difference in developmental potential through the blastula stages of two developmental types of presumptive heterozygotes, clutches could have been divided into groups of non-cellularized, cellularized with developmental defects, and normally developing embryos. The indicated graphs have been replotted as scatter plots with the addition of new data which gave very similar results. In Fig4 and Fig6, we did not separate the eggs according to their cellularization status because they were taken at the one-cell stage (<1 hpf), thus, it was not be possible to determine their developmental status. We harvested the eggs at the earliest stage because we wanted to investigate the maternal effect of the genes, and we considered that there would be substantial changes to the levels of these transcripts if the eggs were harvested later since we assumed that they would be used and possibly degraded during the first few cell divisions.

Description of male transmission of mutated foxr1 genes is lacking or confusing (lines 271 and 364-365). How many F0 males were mated? What were the results? Why no qPCR data on testes, given 2 males were dissected for tissue analysis? In fact, we obtained a lot more F0 males than females since disruption of ovary-specific genes often render more males in zebrafish. Thus, we had mated around 10 F0 males with no mutant progeny. Since we are investigating maternal effect genes, we did not probe further into male transmission and did not investigate testes.

Line 295 should read Fig 5M rather than 4M. This has been corrected.

The gels shown in Figure 5 and in peerj-27280-Uncropped_image_Figure_5 do not appear to be the same exposure of the same gel. The bands look dissimilar. This was our mistake. The uncropped blot was used in a previous version of the manuscript. The uncropped blot that was used for this manuscript with defining labels added has been added.

## Validity of the findings

The Cas9 deletion approach and the ability of the authors to detect fertilization by assaying a transgene only from sperm is a compelling assay for fertilization in non-developing embryos and supports the conclusion of a maternal effect for foxr1.

I am not expertly qualified to comment on the phylogenetic analysis of this paper, but the tree seems like a reasonable fit to the data. Would a second BLAST search for homologs using another species than zebrafish be warranted to validate the results found searching only with zebrafish proteins? We actually performed BLAST search with human and medaka Foxr1 protein as well. This description has been added to the revised manuscript.

I think the authors meant the F1 generation at line 272 since the F0 generation were viable and fertile (excepting foxr1-2) demonstrating no direct phenotype assayed by the authors. This has been corrected.

A speculation at lines 217-218 and again at 330-333 is that the different sequences found in fish comes from a gene duplication event is not supported by the data. This is troublesome given that syntenic analysis is obviously done on one of the Foxr1 protein sequences from gar and medaka etc., but not the other. Given the apparent ability of the authors to compare syntenic regions, an explanation about the other sequences and their synteny is warranted. Only sequence obtained from different databases resulted in unique isoforms of the sequence. When speculating, the authors should provide more than one interpretation, and where possible how one would test between the possibilities (lines 340-342 makes the vague statement "Further analyses on the two copies" without explaining the analyses). We agree that more than one explanation is needed. So, we have added the phrase, " or technical differences due to different sequencing procedures".  We did not perform further syntenic analyses as our goal was to analyze the *foxr1* gene in zebrafish, which harbour just one gene.

The word penetrance (line 276) is used incorrectly in a genetics paper as penetrance is the condition when an phenotype-linked allele is present, but the affected organism does not exhibit a phenotype. The authors had made the point in the paragraph preceding that all surviving embryos were WT. This indicates that the F0 generation were mosaic and many eggs did not have foxr1 disruption. I suggest the word frequency instead.  This has been corrected.

A comment on the differences seen between qPCR levels and RNA-seq levels in bone is warranted.   The difference in the level of *foxr1* in bone could be due to methodological differences; RNA-seq is more sensitive than qPCR since it is designed to detect the sequences all along the transcript while the latter detects just one area of the transcript. Thus, it is possible that a different splice variant of *foxr1* exists in bone. We added the following sentence to the revised manuscript, " A high expression of *foxr1* transcript was also detected in bone by RNA-seq, but not by qPCR; this could be due to methodological differences as RNA-seq is more sensitive than qPCR since it is designed to detect the sequences all along the transcript while the latter detects just one area of the transcript. Thus, it is possible that a different splice variant of *foxr1* exists in bone."

The sentence of lines 387-388 is confusing with verb subject disagreement.  This has been corrected.

The expression claim in lines 388-389 needs citation.  This has been corrected.

The conclusion statement on 393 and repeated at 399 "via the p21 and mTOR pathways" has been correlated, but not tested in this paper. We modified the conclusion statement to the following, "Our findings suggest that maternally-inherited *foxr1* may be required for the first few cleavages after fertilization for proper cell growth and proliferation possibly via *p21* and *rictor*, since deficiency in *foxr1* leads to either complete lack of or abnormal cell division culminating to early death in the fertilized egg."

Opossum Foxr2 seems given the wrong color in Figure 1. Should it be blue like Armadillo? The opossum is foxr1, not foxr2, so the color is correct.

# Reviewer 2 (Anonymous)

## Basic reporting

This is a well-designed and described study from a prominent fish reproduction biologist. The major findings included:
The presence of foxr1 in a number of species.
In zebrafish, foxr1 has predominately ovarian specific expression.
In zebrafish oocytes, foxr1 is expressed early and accumulates through oogenesis.
When foxr1 is knocked out, oocyte are still fertilized but have problems undergoing normal cell division. This is due to an increase in p21 and a decrease in rictor.

I thought this was a well written paper with a good depth to the introduction and very straightforward study design and results:
Minor issues
A few awkward or confusing sentences:
*note: my complete review download has different line numbers than the individual paper download.
We have modified the following sentences to improve the text as much as possible.

Line 43: In vertebrates, maternal products including transcripts, proteins, and other biomolecules are necessary for kick-starting early embryonic development until the mid-blastula transition (MBT) when the zygotic genome is activated [1].
This sentence has been modified to, " In vertebrates, maternal products including transcripts, proteins, and other biomolecules are necessary for initiating early embryonic development from fertilization until the mid-blastula transition (MBT) when the zygotic genome is activated."

Line58: foxl2 and its relatives are known to be key players in ovarian differentiation and oogenesis in vertebrates; it is essential for mammalian ovarian maintenance and through knockout experiments, it was demonstrated that foxl2is a critical regulator of sex determination by regulating ovary development and maintenance also in Nile tilapia, medaka, and zebrafish[9]
This sentence has been modified to, "*foxl2* and its relatives are known to be key players in ovarian differentiation and oogenesis in vertebrates. Foxl2 is essential for mammalian ovarian maintenance, and it was demonstrated in Nile tilapia, medaka, and zebrafish that *foxl2* is also a critical regulator of ovary development and maintenance."

Line 333: It is also possible that foxr1 was duplicated in the ancestral actinopterygii and subsequent gene losses in bowfin as well as in the teleosts especially following the multiple gene duplication events such as the teleost-specific whole genome duplication (TGD) and salmonid-specific whole genome duplication (SSGD).

This sentence and one following it have been modified to, " It is also possible that *foxr1* was duplicated in the ancestral actinopterygii followed by the loss of 1 copy in a lineage-dependent manner. Finally, it appeared that the different whole-genome duplication events, the teleost-specific genome duplication (TGD) and salmonid-specific genome duplication (SaGD) did not impact the current *foxr1* gene diversity because in most species, only one *foxr1* gene was retained."

Line 345: Previous reports have demonstrated the predominant expression of foxr1 mRNA in the ovary of medaka, eel, and tilapia[8,11,13], but in the male germ cells and spermatids in mouse and human[24].

This sentence has been modified to, "Previous reports have demonstrated the predominant expression of *foxr1* mRNA in the ovary of medaka, eel, and tilapia[8,11,13], but it was found mostly in the male germ cells and spermatids in mouse and human."

Vasa::gfp line wasn't described when it was first mentioned, but was later in the text. Helpful to know why this was used earlier for people outside of the field of maternal-effect genes
We modified the Materials and Methods to include a short description of this gene, "Once confirmed, the mutant females were mated with males harboring the *vasa::gfp* gene, where *vasa* was fused with *gfp* at the 3' end, to produce F1 embryos.....". *vasa* is a gene that is specific for germline progenitor cells that are present in both males and females, so it is not specific for maternal-effect genes. The mutant we used was created by Dr. Jean-Jacque Lareyre who works on germline progenitor cells.

Problem with citation 4
Years missing in citations 9 and 11 This has been corrected.

## Experimental design

Well designed. I have a few minor comments:
No section in the methods on animal husbandry. This has been added.

npm2b mutants are mentioned on Line 288 but without any background as to what or why they were mentioned. This extends to Fig. 5M We wanted to draw parallels between *foxr1*, a potential maternal-effect gene, and *npm2b*, a known maternal-effect gene. Since *npm2b* has been shown to be extremely highly expressed in maturing and mature eggs, we believed that it was a good control. We have added a discussion of this in the Discussion section.

Tissue samples came from 2 males and 3 females. It is not clear why only 2 males were used. Aren't you treating them as separate groups since you are looking at a sex specific expected phenotype? Furthermore, the way they are reported in Fig. 3, it appears that they are just grouped together. Clarity in the description in the methods would probably clarify the design better and make more clear Fig. 3. Furthermore in the legend for Fig. 3, it says that tissues were harvested from 3 to 4 WT zebrafish? At first, we collected samples from males and females to see if there were sex-specific differences, but there were none between the sexes. Thus, we grouped them together as just "wildtype tissue". We used 3-4 WT zebrafish because some of the tissues were different to harvest or to obtain RNA so we could not

examine RNA from all of the fish.

Fig. 4C: there are no WT controls to compare this information to. You seems to have this information for 4B, therefore it should be added to Fig.4C for comparison.  This has been added.

Fig. 4C the cross indicates that all zebrafish in these groups showed the exact same phenotype of a blastodisc sitting atop an enlarged syncytium? Or is this just an example of a developmental defect. Clarity would be good for this.  The only observed developmental defect was unsmooth and irregularly-shaped yolk with a blastodisc sitting on top of an enlarged syncytium. We did not observe other types of defects.

Fig. 5M. Why was npm2b used as a control. This isn't clear since it is also missing from the paper.  In this experiment, we needed to examine another gene to show that the genotyping PCR was working well. We chose to use to detect the *npm2b* because we knew that these primers work very well  for genotyping PCR of one-cell eggs, a procedure which is rather difficult due to the very small amount of DNA and the high yolk content.

## Validity of the findings

I am concerned about the uncropped blot. I can't tell what I am looking at in the cropped version (compared to the uncropped blot). I think just a label on the uncropped blot would allow reassurance.  The uncropped blot with the labels has been added.

# Reviewer 3 (Jennifer Liang)

## Basic reporting

The manuscript "foxr1 is a novel maternal-effect gene in fish that regulates embryonic cell growth via p21 and rictor" by Cheung et al. identifies a function for the transcription factor foxr1 in oogenesis and early embryogenesis. They demonstrate that foxr1 mRNA is provided maternally, consistent with this role. Further, through loss-of-function studies using Crispr-Cas9, they find that the number of normal embryos produced from a foxr1 Crispr-Cas9 edited female and a male WT at the foxr1 locus (carrying the vasa::gfp transgene, which should be expressed in the fertilized F1 progeny) is significantly reduced. Phenotypes of the progeny include failure to undergo any cleavages and abnormal cleavages, both of which result in death of the embryos by 24 hours post spawning. As expected from a mosaic parent, a proportion of the embryos are normal. Overall, this is a well done study that will be of interest to the scientific community, and especially to those interested in maternal effect genes and early development. However, there are some changes needed before the manuscript is ready for publication.

## Experimental design

The experimental designs are strong. However, in a few places, additional information is

needed.

A. How many biological replicates were done for the RNA-seq experiments? The RNA-seq experiment was not performed in this study, we simply datamined the publicly open database for results on *foxr1*. The detailed protocol is available in Pasquier et al. *BMC Genomics*, 201, 2016; 17 : 368.

B. For the GFP PCR on F1 progeny with no cell divisions (Figure 5), were the PCRs done on single embryos or groups? This may affect the interpretation of the results. The PCRs were performed on pooled eggs of 50-200 because the egg were collected at the one-cell stage so single embryo PCR would be too difficult due to the small amount of DNA and high amount of yolk.

## Validity of the findings

Major concerns.

1. The claim that Foxr1 functions through regulation of p21 and rictor is not supported by the data in this manuscript. The authors demonstrate that expression of these two genes is altered in the F1 progeny of the foxr1 Crispr-Cas9 edited female X vasa::gfp male. There is no functional data to demonstrate that these changes in gene expression are causing the observed phenotypes in the progeny. The authors should change their title to focus only on their research on Foxr1, they should take the p21 and rictor reference out of the title. There also several places in the Discussion that need to be revised to make it clear that Foxr1 acting through p21 and rictor is a hypothesis or working model only. These places are marked in the manuscript pdf.

We agree with this comment. The title and discussion have been modified accordingly.

2. The Results and Discussion sections are somewhat redundant with each other. The authors should follow the conventions of these sections and make sure text is in the appropriate section. We modified the revised manuscript according to your recommendations.

Minor concerns

1. In several places, the figure legends are not complete or need clarification. These places are marked on the pdf of the manuscript.

We modified the revised manuscript according to your recommendations. In Fig 2, we rechecked the light blue areas and found that they were errors, so we corrected them and changed them to light grey which denotes lack of synteny.

2. Similarly, there are several places in the text that need to be revised to improved clarity. These places are also marked in the pdf of the manuscript.

We modified the revised manuscript according to your recommendations.